# Sustainable Surface Water Storage Development Pathways and Acceptable Limits for River Basins

**Nishadi Eriyagama** [1,*] **, Vladimir Smakhtin** [2] **and Lakshika Udamulla** [3]

1   International Water Management Institute (IWMI), P.O. Box 2075, Colombo 10120, Sri Lanka
2   UNU Institute for Water Environment and Health, 204–175, Longwood Road South, Hamilton, ON L8P 0A1, Canada; vladimir.smakhtin@unu.edu
3   Faculty of Engineering Technology, Open University of Sri Lanka, P. O. Box 21, Nawala, Nugegoda 10250, Sri Lanka; laudu@ou.ac.lk
*   Correspondence: n.eriyagama@cgiar.org

**Abstract:** This paper addresses the questions of acceptable upper limits for storage development and how best to deploy storage capacity in the long-term planning of built surface water storage in river basins. Storage-yield curves are used to establish sustainable storage development pathways and limits for a basin under a range of environmental flow release scenarios. Optimal storage distribution at a sub-basin level, which complies with an identified storage development pathway, can also be estimated. Two new indices are introduced—Water Supply Sustainability and Environmental Flow Sustainability—to help decide which pathways and management strategies are the most appropriate for a basin. Average pathways and conservative and maximum storage limits are illustrated for two example basins. Conservative and maximum withdrawal limits from storage are in the range of 45–50% and 60–65% of the mean annual runoff. The approach can compare the current level of basin storage with an identified pathway and indicate which parts of a basin are over- or under-exploited. A global storage–yield–reliability relationship may also be developed using statistics of annual basin precipitation to facilitate water resource planning in ungauged basins.

**Keywords:** surface water storage; reservoirs; sustainability; environmental flows; acceptable limits; pathways

## 1. Introduction

Storing water helps in stabilizing variable water supplies, producing hydropower, and mitigating floods. Water storage options include small and large reservoirs, ponds and tanks, aquifers, soil moisture, and wetlands [1]. Reservoirs, ponds, tanks, and wetlands are generally regarded as surface storage, and the others as subsurface storage. Surface storage has been beneficially exploited for thousands of years. Examples span from dams constructed by Sumerians in the Euphrates and Tigris Rivers in 6500 BC [2] and the ancient tank irrigation systems that flourished in Sri Lanka from around 300 BC [3,4] to the mega dams of the twentieth century in different parts of the world [5–7]. However, surface storage is known to create adverse ecological and social consequences, too.

At least 43% of all global water abstractions for irrigation are supplied by surface storage (reservoirs with a storage capacity larger than 0.1 km$^3$). The remainder is split between groundwater supply (22%), aqueduct transfers (8%), and other unspecified sources (27%) [8]. Hydropower accounts for close to 17% of global electricity generation, mostly produced by large dams [9,10]. Reservoirs support a number of other non-consumptive activities such as flood risk reduction, recreation, fishing, and small-scale livelihoods. Apart from the tangible benefits, in certain countries, large-scale water storage infrastructure serves as a strategic tool for promoting domestic peace and stability [11,12] and a symbol of national identity and patriotism [13]. Large water storage infrastructure combined with

transboundary water transfers also function as a foreign policy tool to enhance influence at the international level [11,14,15]).

On the other hand, only 23% of rivers longer than 1000 km remain unfragmented by dams [16]. Dams regulate river flow [17,18] and alter downstream flow regimes, which negatively impacts aquatic ecosystems [16,19–21] and in turn fisheries, livelihoods, and cultural and recreational activities [22–24]. Large dams often displace a significant number of people and affect those living downstream, causing losses to property, livelihoods, and incomes [25–27]. The number of people displaced by dams in the last century is at least 80 million [28]. Furthermore, construction of large dams frequently creates conflicts between competing expectations of different groups of stakeholders [15,29].

The current global tally of large dams (15 m high or greater, and between 5 m and 15 m impounding more than 3 million m$^3$) is close to 60,000 and growing [30]. An indicative figure for the number of smaller dams below the International Commission on Large Dams (ICOLD) benchmark is 800,000 and also growing [31]. While surface water storage capacity continues to grow globally, its development remains largely ad-hoc, i.e., driven by the needs of the immediate future, the availability of suitable sites, and short-term political interests. The question of what is an acceptable upper limit for surface storage development in a river basin remains unanswered. Concepts of closed basins [32], sustainability boundaries [33], and water footprints [34] prescribe limits to withdrawals from available river flow but do not suggest an acceptable or optimal storage limit for a particular river basin. Related questions such as how best to deploy the maximum acceptable storage capacity in a river basin in terms of the size, number, and location of reservoirs that maximizes benefits and minimizes environmental and social costs have not been well studied. A few studies, such as [35] and [36], have dealt with the latter question, but not in the context of an upper limit to surface storage development.

There is a significant body of literature on the optimal operation of particular reservoirs or reservoir systems to meet human and ecological water requirements rather than considering them in a basin planning context. Examples include studies [37–41], which seek to optimize operations of existing reservoirs to meet multiple objectives ranging from water supply, flood control, energy generation, and environmental flow. Although there are numerous other examples, a lengthy discussion is beyond the scope of this paper.

This paper moves beyond traditional optimization of existing reservoir operations towards a long-term, multi-decadal vision for surface storage development in a river basin. It recognizes that an acceptable upper limit to storage development is a compromise between storage benefits and ecosystem and social costs [42] and that tools to support the decision-making process for structured storage development are needed. The paper examines how best to determine this compromise between storage benefits and ecosystem costs when developing storage for supplying water for various human needs. It investigates the behavior of a set of indicators measuring water supply benefits and environmental flow releases at different storage capacities, numbers, modes of siting, and spatial distribution of reservoirs (different scenarios of reservoir system configurations).

The paper illustrates how an overarching sustainable storage development pathway and acceptable limits to surface storage development may be identified for a river basin to limit withdrawals from reservoir storage to a sustainable level. It also illustrates the optimal manner in which storage capacity and sustainable withdrawal levels may be distributed among sub-basins so that a river basin as a whole complies with the overarching sustainable storage development pathway, even when the reservoir configuration changes. Finally, it discusses how the results may be used in practice to identify drainage regions having higher surface storage potential, to plan for a more beneficial distribution of storage capacity, and to allocate environmental flows (EF).

## 2. Materials and Methods

### 2.1. Example Case Study Basins

Approaches presented here can be applied to any river basin. For illustrative purposes, this paper uses two case study basins in Sri Lanka: the Malwatu Oya in the Dry Zone and the Kalu Ganga in the Wet Zone (Figure 1). The Dry Zone (stretching to the north and the east of the island) has a mean annual precipitation (MAP) of less than 1750 mm and a pronounced dry season with high evaporation rates from May to September. The Wet Zone lies in the southwestern part of the country and receives a MAP of over 2500 mm, which is normally spread throughout the year. This zoning is used in Sri Lanka, which is largely humid overall, hence, the characterizations "wet" and "dry" are relative to the national context. Apart from different climatic zones, the two basins contrast in terms of terrain, existing storage infrastructure, basin area and shape, type of land cover and land use, and dominant water issues.

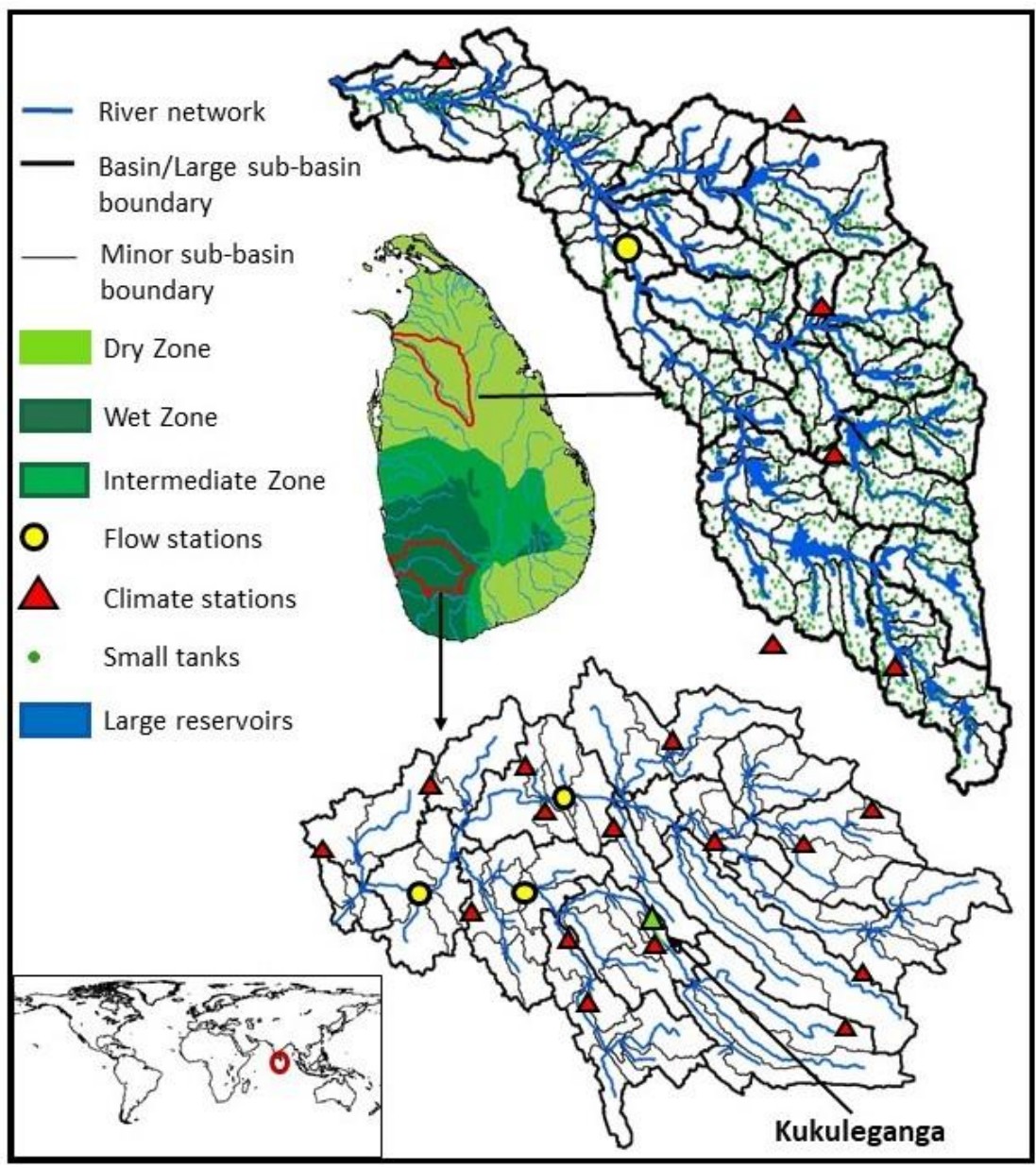

**Figure 1.** Study basins in climatic zones of Sri Lanka (**center**), the sub-basin structure, and hydrography of the Malwatu Oya (**top**), and the Kalu Ganga (**bottom**) basins.

The Malwatu Oya basin has a drainage area of 3338 km$^2$, a MAP of 1300 mm, and a natural mean annual runoff (MAR) of 0.79 km$^3$ (coefficient of variation 0.47). Its elevation varies from 0 to 740 m amsl. Its gently undulating terrain is dotted with small tank cascade systems that originated in the Third Century BC (Figure 1). This system consists of a series of interconnected small reservoirs that collectively store, regulate, and release incoming flow to paddy cultivations [3]. Apart from 1853 small tanks, this basin has several larger irrigation reservoirs, also built in ancient times but subsequently modified to suit the present-day demands of paddy cultivation. The combination of small and large reservoirs (which store about 43% of the MAR) strives to satisfy the irrigation and domestic needs of the basin during the long dry period from May to September. However, intense water demand and frequent droughts have led to water scarcity in the basin [43].

The Kalu Ganga basin, covering an area of 2296 km$^2$, is about 30% smaller than Malwatu Oya, with a MAP of about 4000 mm. While the natural MAR of Kalu Ganga (8.45 km$^3$) is more than 10 times that of Malwatu Oya, its coefficient of variation of annual flow (0.14) is much less. The basin's elevation varies from 0 to 2250 m amsl and has a steep gradient in the upper catchment area, but milder low-lying slopes further downstream. Basin agriculture consists mainly of perennial rain-fed crops (as opposed to irrigated paddy in Malwatu Oya) such as tea and rubber. Flooding occurs in the low-lying areas of the basin almost every year [44]. Currently, there is only one human-made reservoir within the basin (Kukuleganga, a run-of-the-river hydropower reservoir storing only 0.02% of the MAR), but there are several proposals for more reservoirs to harness the abundant water resources here for inter-basin transfer and for developing hydropower [45].

### 2.2. Data and Reservoir System Configuration Scenarios

The main data used in the analyses are simulated time series of natural monthly flows for the period 1961–2013 at the outlets of 148 and 114 minor sub-basins of the Malwatu Oya and Kalu Ganga basins, respectively (Figure 1). Natural or unregulated flows—flows that would occur at a particular location of a river in the absence of any upstream water resources development—were simulated by using the Soil and Water Assessment Tool (SWAT) model. Figure 1 shows the locations of climate stations (6 for Malwatu Oya and 15 for Kalu Ganga, which recorded data for periods ranging from 1948 to 2013) that were used as inputs to the two models. The SWAT model for Malwatu Oya was calibrated and validated for the periods 1971–1975 and 1976–1979, respectively, against observed flows at a heavily regulated location (the only record available; Figure 1). The calibration efficiency was 0.89 for R$^2$ and 0.86 for the Nash–Sutcliffe Coefficient (NSC), while the validation efficiency was 0.82 for R$^2$ and 0.90 for the NSC. The Kalu Ganga model was calibrated and validated for the same periods against three gauging stations (2 regulated and 1 unregulated; Figure 1), obtaining calibration efficiencies of 0.90 for R$^2$ and 0.87 for the NSC, and validation efficiencies of 0.83 for R$^2$ and 0.88 for the NSC.

Subsequently, all existing reservoirs and water management practices were removed from the model setup and the models were executed under current land cover and land use conditions. The output flow time series from the modified models were assumed to represent pseudo-natural or at least unregulated flow. It is acknowledged that the validity of these simulated flows depends on having a high number of sites with observed flow data (both regulated and unregulated) against which to compare model results. However, since these pseudo-natural flows are used to illustrate a generic approach for long-term water storage planning rather than obtain results specific to the study basins, the limitations in the simulated flow time series are of little significance. The SWAT models only play a support function for developing generic methodologies (Sections 2.3–2.6), which may be implemented in any other basin using relevant input data.

To develop generic methodologies, this research analyzed three hypothetical reservoir system configuration scenarios (Figures 2 and 3). Configuration 1 consists of a single reservoir placed at the downstream end of the basin through which all runoff is routed. In Configuration 2, hypothetical reservoirs are placed near the outlets of 15 and 17 large

sub-basins (those recognized by the Department of Agrarian Development) of the Malwatu Oya and Kalu Ganga basins, respectively. In Configuration 3, the sub-basins are further disaggregated into 148 and 114 minor sub-basins of the Malwatu Oya and Kalu Ganga basins, and hypothetical reservoirs are placed near their outlets. The main features of each configuration are presented in Table 1. The sizes of individual reservoirs decrease gradually from Configuration 1 to Configuration 3.

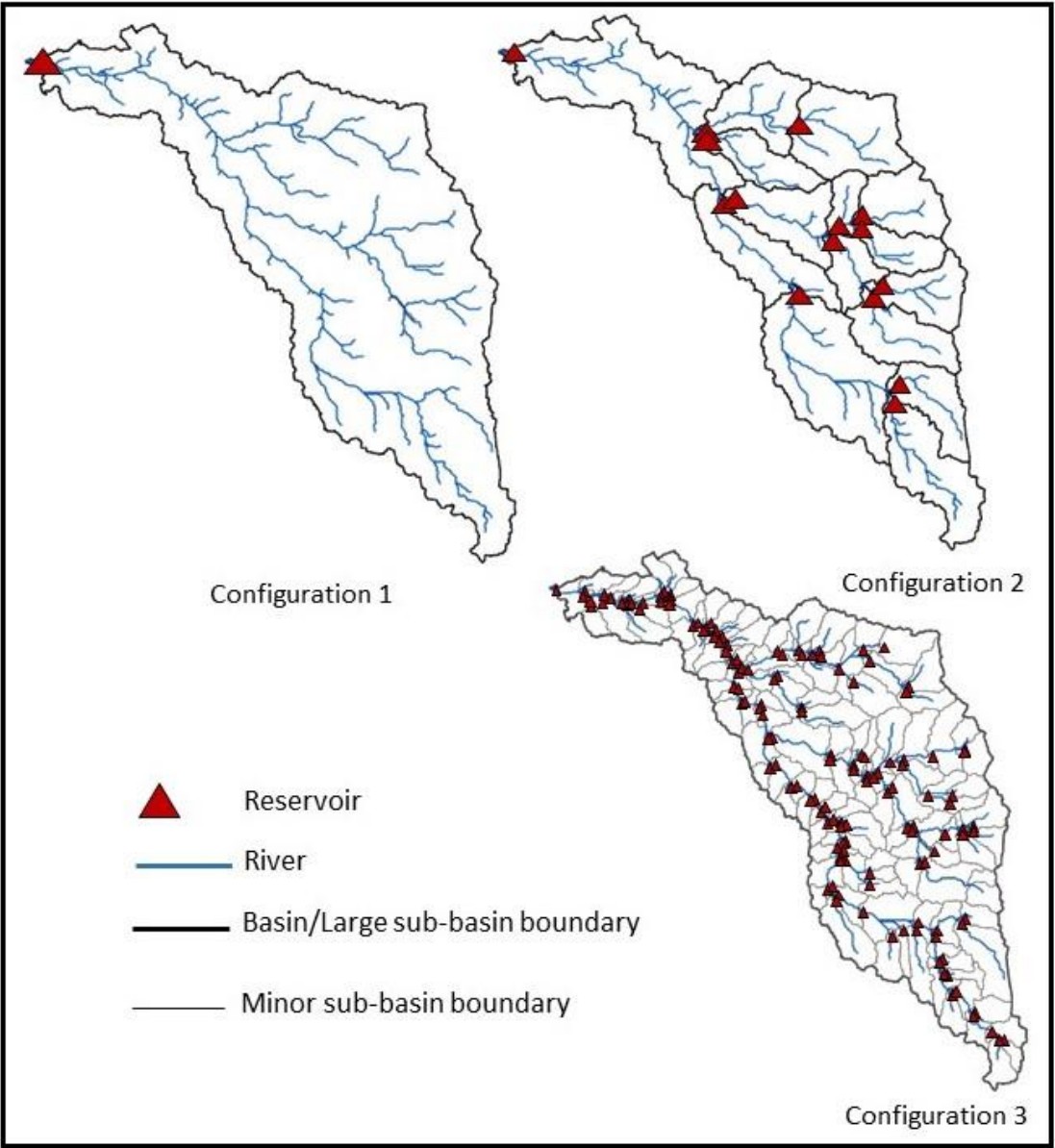

**Figure 2.** Three reservoir system configurations in Malwatu Oya ranging from one large hypothetical reservoir at the downstream end to 15 medium-size reservoirs in large sub-basins and 148 smaller reservoirs in minor sub-basins.

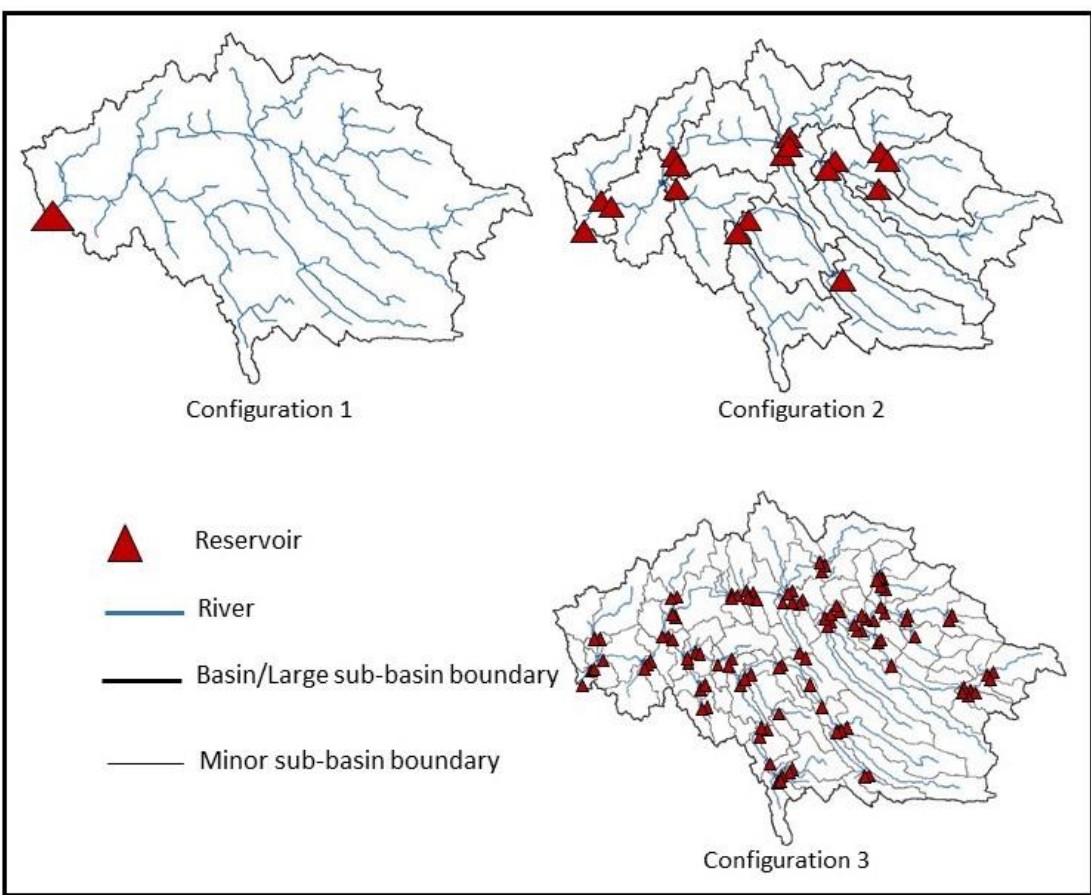

**Figure 3.** Three reservoir system configurations in Kalu Ganga ranging from one large hypothetical reservoir at the downstream end to 17 medium-size reservoirs in large sub-basins and 114 smaller reservoirs in minor sub-basins.

**Table 1.** Main features of reservoir system Configurations 1, 2, and 3.

| Basin | Malwatu Oya | | | Kalu Ganga | | |
|---|---|---|---|---|---|---|
| Configuration | 1 | 2 | 3 | 1 | 2 | 3 |
| Number of reservoirs | 1 | 15 | 148 | 1 | 17 | 114 |
| Average catchment area per reservoir (km²) | 3338 | 223 | 22.6 | 2296 | 172 | 25.7 |
| Average catchment area as a fraction of total area | 1 | 0.07 | 0.007 | 1 | 0.06 | 0.009 |
| MAR of average catchment area (km³) | 0.79 | 0.06 | 0.006 | 8.45 | 0.51 | 0.075 |
| MAR of average catchment area as a fraction of total MAR | 1 | 0.07 | 0.007 | 1 | 0.06 | 0.009 |

Note: MAR stands for Mean Annual Runoff.

*2.3. Overarching Surface Water Storage Development Pathways*

In this paper, the term "water supply" means the supply of water to all consumptive uses including irrigation, industrial, domestic, and other purposes, which together make up a major portion of the water use portfolio of a river basin. We limit our focus to the water supply at this stage to simplify the analysis. The approach presented in this paper may be extended or adjusted to cover other non-consumptive water resource uses.

The economic benefit from a water supply reservoir may be measured in terms of the annual reservoir yield. This is an estimate of how much water can be supplied annually for

all those purposes from a reservoir of a given capacity at a given reliability of supply [46] and is usually determined by analyzing the time series of inflows at the location of the reservoir. The annual yield that can be supplied at 100% reliability, for a given flow record by a reservoir of given capacity, which starts full and refills at least once after the worst drought on record, is termed the Safe Yield [47]. This paper refers to Safe Yield as the Water Supply (WS) Yield.

Storage-WS Yield curves (graph of WS Yield with increasing storage capacity) are usually used to design single reservoirs. However, in this research, Storage-WS Yield curves were used as a tool to illustrate the design of overarching surface water storage development pathways for an entire river basin under seven environmental flow (EF) release scenarios, by assuming that a single large reservoir is placed at the outlet of the entire river (reservoir system Configuration 1; Figures 2 and 3).

2.3.1. Environmental Flow Release Scenarios

The seven EF release scenarios consisted of (a) a zero-release scenario where no specific EF releases were allowed except spills from the reservoir, and (b) six EF release scenarios generated by using the method of Smakhtin and Anputhas [48]. This method uses natural or unregulated monthly flow time series at a particular location as input to generate six other monthly flow time series which differ in magnitude but are similar in pattern to the natural flows. The method first estimates a flow duration curve of the natural flows, which is then shifted to the left along the horizontal axis to estimate six environmental flow duration curves that are subsequently converted to time series of environmental flows. These flow time series correspond to six Environmental Management Classes (EMCs), named F through A. The annual and monthly quantities of flow releases increase in magnitude when moving from Class F to Class A. Class F represents the ecological status of a highly degraded river, whereas Class A represents that of a protected healthy river (see [48,49] for further details). When reservoir system Configuration 1 is assumed, these time series of flows represent monthly EF requirements at the most downstream end of the river to maintain it in each of the EMCs. Figure 4 shows extracts of the monthly time series of EF release requirements (corresponding to the six EMCs) for the Malwatu Oya and Kalu Ganga basins that were generated in this manner starting with the natural flows.

In a similar context to WS Yield, the annual quantity of EF released downstream from a given reservoir is referred to as the EF Yield. Table 2 summarizes the minimum EF Yield (as a percentage of the MAR) expected to be released downstream under each of the seven EF release scenarios considered in this exercise.

**Table 2.** Environmental Flow (EF) Yield requirement at the downstream end of Malwatu Oya and Kalu Ganga basins under seven EF release scenarios.

| EF Release Scenario | Environmental Management Class (EMC) | EF Releases as Percentage of MAR | |
|---|---|---|---|
| | | Malwatu Oya | Kalu Ganga |
| 1 | No EF | 0 | 0 |
| 2 | F | 9.7 | 25 |
| 3 | E | 12.6 | 33.3 |
| 4 | D | 16.6 | 43.1 |
| 5 | C | 23.2 | 54.6 |
| 6 | B | 35.5 | 67.8 |
| 7 | A | 58.3 | 82.8 |

Note: The zero-EF release scenario is shown as EMC No EF.

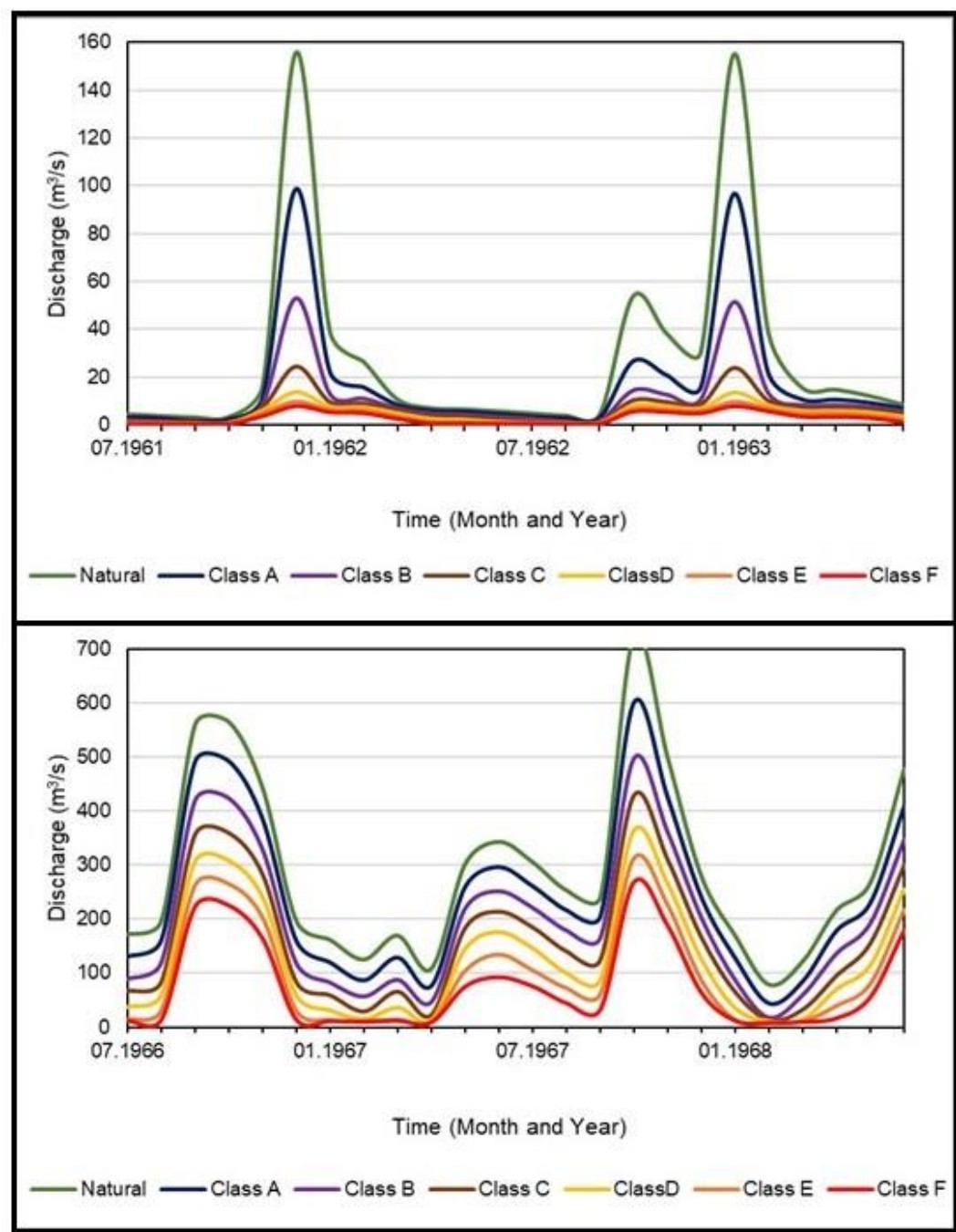

**Figure 4.** Extracts from Environmental Flow time series required at the outlets of Malwatu Oya (**top**) and Kalu Ganga (**bottom**) for the rivers to be maintained in six Environmental Management Classes (EMCs).

### 2.3.2. Reservoir Simulation

The WS Yield corresponding to gradually increasing storage capacities under reservoir system Configuration 1 was estimated with the Sequent Peak Algorithm [50,51] as implemented in the Water Evaluation and Planning (WEAP) model. A varying monthly water demand, closely aligned with the actual temporal distribution of the annual water demand of the Malwatu Oya basin (nearly 65% of which is between October to February, corresponding to the main paddy cultivation season), was imposed on both basins to ensure that the results for the two basins were comparable. For each reservoir capacity, EF releases were varied from zero (hereafter referred to as Class No EF) to Class A so that seven corresponding WS Yield values were obtained. The largest WS Yield was obtained

for Class No EF, whereas the smallest WS Yield was obtained for Class A. A range of seven overarching Storage-WS Yield curves were estimated for each basin in this manner (Figure 5). The term "overarching" is used to imply that these Storage-WS Yield curves are for the entire basin.

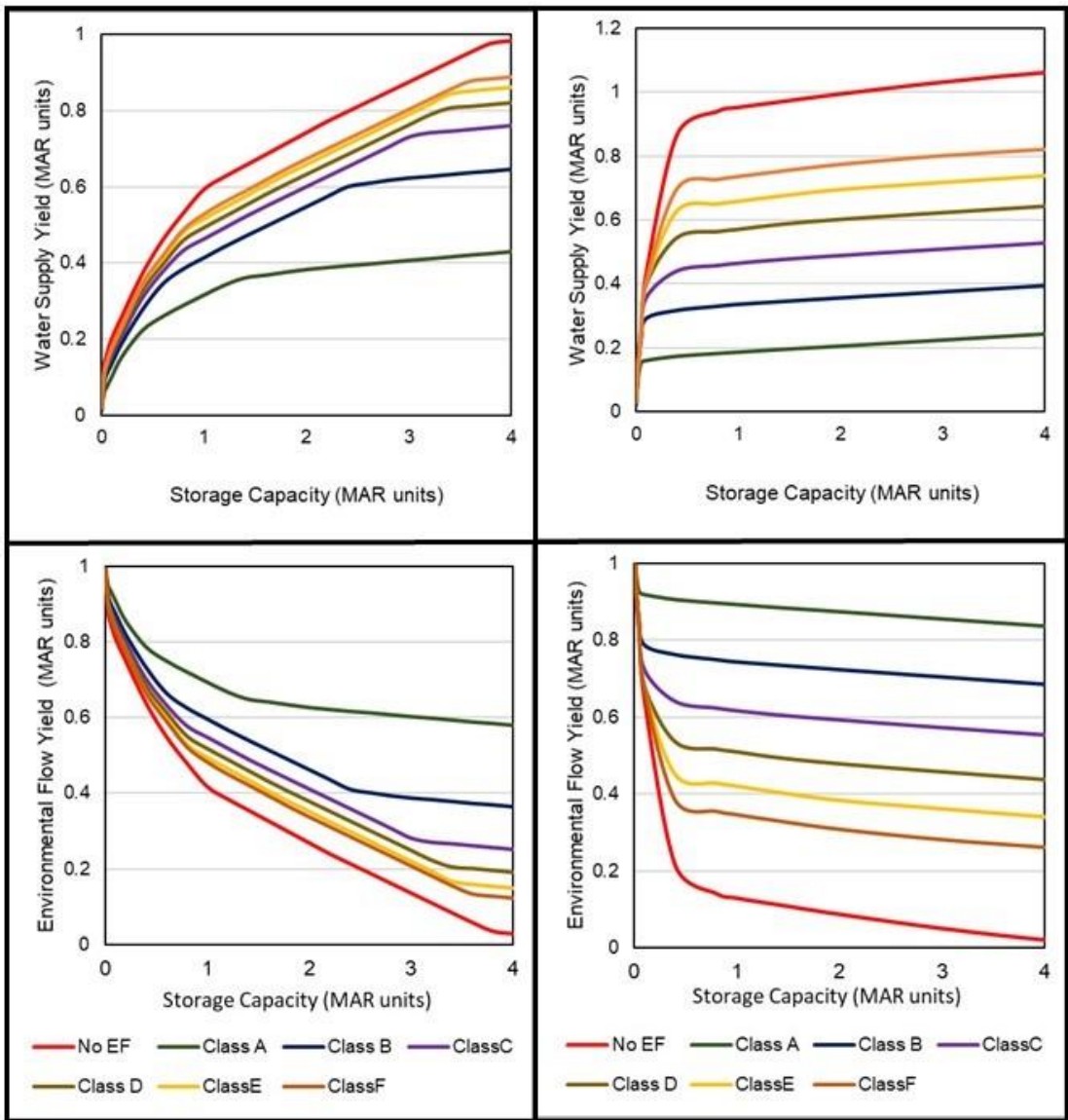

**Figure 5.** Behavior of Water Supply Yield (**top**) and Environmental Flow Yield (**bottom**) under different Environmental Management Classes (EMC) for a hypothetical reservoir through which all river runoff is routed for Malwatu Oya (**left**) and Kalu Ganga (**right**).

The basic equations of the Sequent Peak Algorithm are:
For i= 1 to N and k=1 to 12,

$$S_i = S_{i-1} - \propto \beta_k \mu - EF_{i-1} + Q_{i-1} \text{ If } S_i > 0 \tag{1}$$

Else,

$$S_i = 0 \tag{2}$$

$$S_0 = S_c \tag{3}$$

where $S_i$ = reservoir storage at the beginning of the month i; $Q_{i-1}$ = reservoir inflow during month i−1; $S_{i-1}$ = reservoir storage at the beginning of the month i−1; $\mu$ = mean annual river flow; $\beta_k$ = fraction of annual demand expected in month k (corresponding to month i−1) out of 12 months of the year; $\propto$ = total annual demand as a fraction of the mean annual river flow; $EF_{i-1}$ = environmental flow release during month i−1; N = total number of months in the simulation; $S_0$ = reservoir storage at the beginning of the simulation, and $S_c$ = storage capacity of the reservoir.

The goal of the simulation is to find the maximum $\propto$ for each storage capacity $S_c$ that satisfies the condition $S_0 = S_c$ and $S_i = S_c$ for at least one $S_i$ after the worst drought on record.

Gross WS Yields were estimated assuming zero evaporation or seepage losses from reservoirs. Furthermore, reservoir operation did not consider any conservation, buffer, or inactive storage zones. These assumptions were for simplicity, although the omitted parameters may be included if deemed critical. Reservoir storage was allowed to become zero if it was necessary to satisfy the demands but was required to fill up at least once during the total simulation period (53 years) to ensure continuity of storage. While releasing EF, spills from the reservoir were allowed whenever the reservoir reached its full capacity.

Assuming that all streamflow except the WS Yield flows downstream from the reservoir, a set of seven storage-EF Yield curves were generated for each river (Figure 5).

*2.4. Average Overarching Surface Water Storage Development Pathways*

Average overarching Storage-WS Yield curves of the range of seven curves developed earlier for each basin were estimated by considering approaches (a) and (b) below:

(a)　Considering annual WS Yield

Under this approach, the mean of the two extreme WS Yield values (corresponding to classes No EF and A) for each storage capacity of the reservoir system Configuration 1 was estimated. This resulted in an average overarching Storage-WS Yield curve (hereafter referred to as the Mid curve) for each of the two basins under an average EF Yield scenario.

(b)　Considering monthly variation of WS Yield

This approach investigated ways in which a compromise may be reached between the two extreme Storage-WS Yield curves considering the actual monthly variation of the WS and EF Yields. A WS Sustainability Index and an EF Sustainability Index were formulated following [52] as described below.

1.　WS Sustainability and EF Sustainability

$$Reliability = \frac{N_s}{N} \tag{4}$$

$$Resilience = \frac{N_{us}}{N - N_s} \tag{5}$$

$$Vulnerability = \frac{\sum_{i=1}^{N} m_i}{N - N_s} \tag{6}$$

$$Relative\ Vulnerability = \frac{Vulnerability}{Target\ Demand} \tag{7}$$

$$(WS/EF)\ Sustainability = \frac{Reliability + Resilience + (1 - Relative\ Vulnerability)}{3} \tag{8}$$

where $N_s$ = number of months in which the target demand is met; N = total number of months in the period of analysis; $N_{us}$ = number of months in which the supply has returned to the target demand after failure; and $m_i$ = the magnitude of failure in the ith failure month. The value of both WS Sustainability and EF Sustainability indices varies from 0 to 1.

2.　Compromising WS Sustainability and EF Sustainability

As estimated in Section 2.3, for each size of reservoir there is a range of WS Yields and EF Yields corresponding to different levels of ecological protection for the river (Figure 5). The behavior of WS Sustainability against EF Sustainability for each size of reservoir was investigated assuming: (a) water supply health is at its best (i.e., WS Sustainability = 1) when the WS Yield is at the higher end of the spectrum (EF Yield at Class No EF), and deteriorates with gradually increasing release levels of EF and, (b) the health of the river is at its best (i.e., EF Sustainability = 1) if the WS Yield is at the lower end of the spectrum (EF Yield at Class A), and deteriorates when gradually increasing levels of yields are drawn out. In practice, it is impossible to maintain both WS Sustainability and EF Sustainability at 1 and some compromise must be reached. This compromise may be achieved in several ways, for example, (1) maintaining EF Sustainability at 1 and allowing WS Sustainability to decline, (2) maintaining WS Sustainability at 1 and allowing EF Sustainability to decline, or (iii) allow both EF Sustainability and WS Sustainability to decline so that both requirements are partially met. In the latter instance, if both water supply and environmental flow demands are satisfied in equal proportions, EF Sustainability and WS Sustainability indices will become equal and minimum. The WS Yield at which the two indices become equal and minimum was estimated for gradually increasing storage capacities under reservoir system Configuration 1 in both basins. A second average overarching Storage-WS Yield curve (hereafter the "compromise curve") was constructed for both basins.

### 2.5. Optimum Distribution of Storage Capacity and Withdrawal Thresholds in Sub-Basins

The overarching Storage-WS Yield curves developed in Sections 2.3 and 2.4 provide an estimate of expected cumulative yields from a basin at different cumulative storage capacities. To establish the optimum distribution of this cumulative storage capacity and corresponding optimum yields within individual sub-basins, the impact of distributed reservoir system configurations 2 and 3 (Figures 2 and 3) on the indicators WS Yield and EF Yield were investigated. Under Configurations 2 and 3, Storage-WS Yield curves for each sub-basin (large sub-basins under Configuration 2 and minor sub-basins under Configuration (3)) were generated using simulated natural runoff, without making explicit environmental flow releases other than reservoir spills. The individual curves were aggregated under two sub-configurations to generate a cumulative Storage-WS Yield curve for the entire basin. The two sub-configurations are: (a) each cumulative storage capacity is distributed equally among sub-basins, and (b) each cumulative storage capacity is distributed among sub-basins according to their MAR ratio. The intention was to identify the sub-configuration that produced the maximum aggregated WS Yield for the entire basin.

### 2.6. Regionalized Storage-Yield Curves

All Storage-WS Yield curves were generated by estimating yields at 11 selected storage capacities (0.0, 0.05, 0.1, 0.4, 0.8, 1, 1.4, 2, 3, 4, and 5 MAR units). These were selected to illustrate the shape of the entire curve, including the initial segment and where it levels off. A factor analysis by principal components of all Storage-WS Yield curves of the 148 and 114 minor sub-basins of the Malwatu Oya and Kalu Ganga basins was undertaken to regionalize the curves and identify the factors underlying their shapes.

## 3. Results and Discussion

### 3.1. Overarching Surface Water Storage Development Pathways

The Storage-WS Yield curves for both basins under the seven EF release scenarios are shown in Figure 5 (top) and the Storage-EF Yield curves under each EMC in Figure 5 (bottom). The curves have been normalized by dividing both axes by the MAR of each basin. These curves represent different overarching surface storage development pathways that may be adopted in the case study basins. The Storage-WS Yield curves illustrate the theoretical upper thresholds for cumulative annual surface water withdrawals in MAR units corresponding to a given cumulative storage capacity in the two basins under a range of ecological protection categories, from No EF releases to a stringent Environmental Man-

agement Class A. The corresponding Storage-EF Yield curves (Figure 5 bottom) illustrate the plausible annual EF releases under each surface storage development pathway.

Similar curves may be developed for other river basins. Any of the curves may be adopted as a potential storage development pathway for an entire river basin. However, the selected pathway will become a sustainable surface water storage development pathway only if it corresponds to the required level of ecosystem protection for a given river. For rivers for which such specific levels of ecosystem protection needs have not been identified, an average storage development pathway may be adopted as explained in Section 3.2.

### 3.2. Average Overarching Surface Water Storage Development Pathways and Acceptable Limits

Average overarching surface water storage development pathways were developed following the two approaches in Section 2.4. While a detailed discussion on the results of approach (a) is not required, the results of approach (b) are discussed in detail in this section.

Figure 6 illustrates the WS Sustainability versus EF Sustainability space for a reservoir of storage capacity with 1.0 MAR units under Configuration 1. The graphs at the top show the behavior of the WS Sustainability and EF Sustainability indices when an attempt is made to release gradually increasing EF Yields starting from its lowest level at Class No EF (where WS Sustainability=1). The graphs at the bottom show the behavior of the two indices when an attempt is made to draw out gradually increasing WS Yields starting from its lowest level at Class A (where EF Sustainability = 1). The arrows illustrate different pathways through which a compromise between the two sustainability indices may be made. Each path corresponds to a unique reservoir management strategy (in this case an entire river basin) and defines the priority assigned to meeting WS demands, meeting EF demands, and storing water (Table 3). On the path Priority: EF, the value of the EF Sustainability index is maintained at 1 while that of the WS Sustainability index is allowed to decline. On the path Priority: WS and Water Storage, the value of the WS Sustainability Index is maintained at 1 while that of the EF Sustainability index is allowed to decline. The path Priority: Equal EF and WS maintains two equal indices (but less than 1) by sharing water deficits in equal proportions between WS and EF requirements. The Compromise pathway was constructed by estimating the WS Yield at which the two indices reach a minimum on the path Priority: Equal EF and WS for gradually increasing storage capacities (Figure 7).

**Table 3.** Reservoir and river basin management strategies through which water supply (WS) and environmental flow (EF) demands can be fully or partially met (see Figure 6).

| Path (Reservoir/River Basin Management Strategy) | Explanation |
|---|---|
| Priority: EF | EF releases are made first before satisfying WS demands; any excess water is stored in the reservoir/river basin after satisfying both demands. |
| Priority: Equal EF and WS | Equal and higher priority is assigned to meeting both EF and WS demands than storing in the reservoir/river basin. Shortages are shared in equal proportion between the two demands. Water is stored in the reservoir/river basin only after meeting both demands. |
| Priority: WS | WS demands are met first, EF releases next, and excess water is stored in the reservoir/river basin after satisfying both demands. |
| Priority: WS and Water Storage | WS demands are met first, storage is allowed next; EF needs are only satisfied through storage spills. |

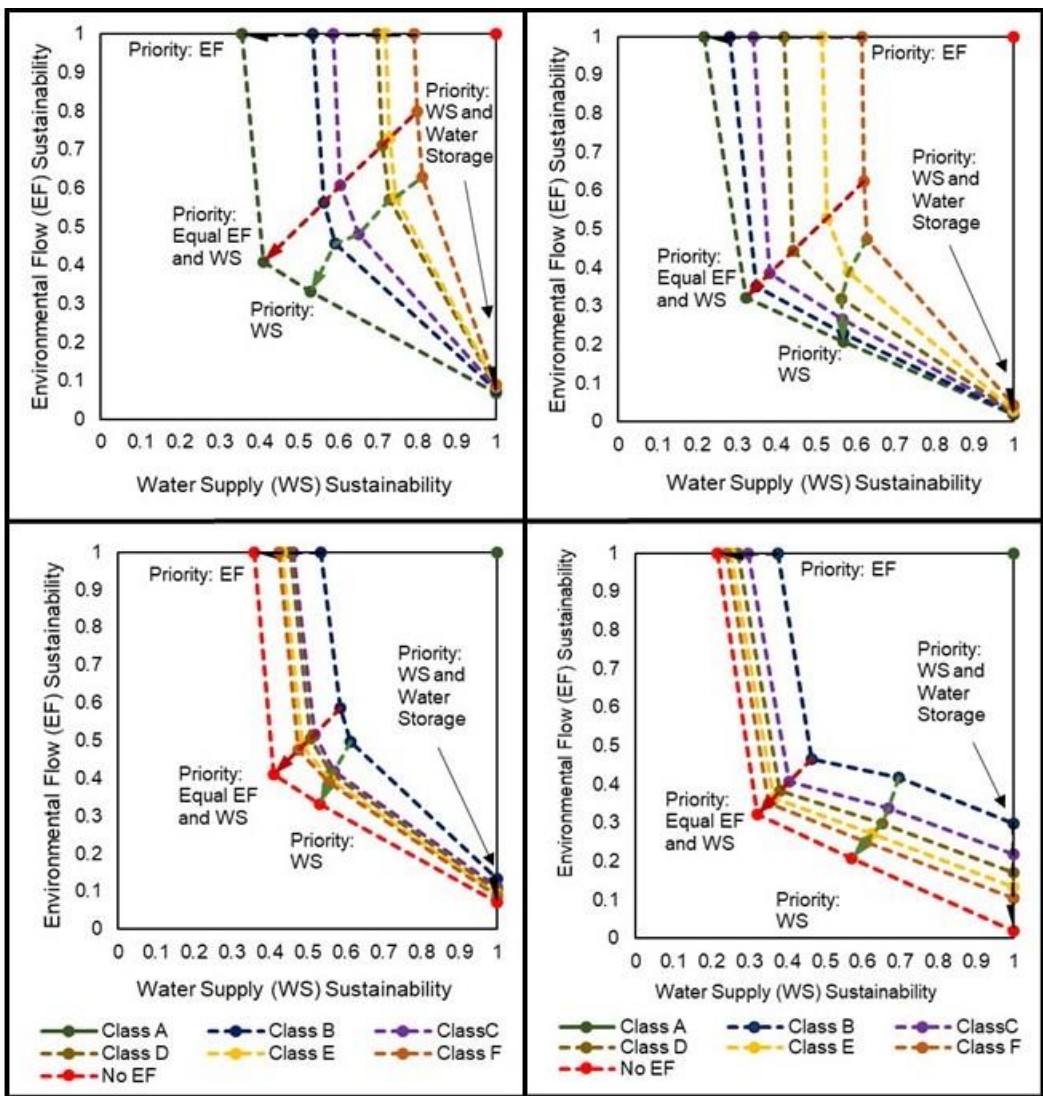

**Figure 6.** Variation of WS Sustainability and EF Sustainability for Malwatu Oya (**left**) and Kalu Ganga (**right**) basins when gradually increasing levels of EF are released (**top**) and gradually increasing levels of yields are withdrawn (**bottom**) from a reservoir with a storage capacity of 1.0 MAR units, with different priorities assigned to water supply, environmental flow provision, and water storage.

Along the fourth pathway, Priority: WS, values of both the EF Sustainability and WS Sustainability indices are less than 1, but the value of the WS Sustainability index is always higher than the EF Sustainability index since water supply demands are prioritized over EF demands and filling the reservoir. Although not exactly an average pathway, it is possible to construct another Compromise pathway (Compromise WS) by estimating the WS Yield at which the two indices reach a minimum on the path Priority: WS. In this case, meeting WS demands are prioritized over meeting EF demands. The exact coordinates of where a river basin is located in Figure 6 depend on the water supply demands, the level of ecological protection required by the river, and the river basin management strategy adopted in terms of priorities assigned to WS demands, EF demands, and reservoir filling. A Compromise pathway appropriate for the river in question can be formulated based on these coordinates.

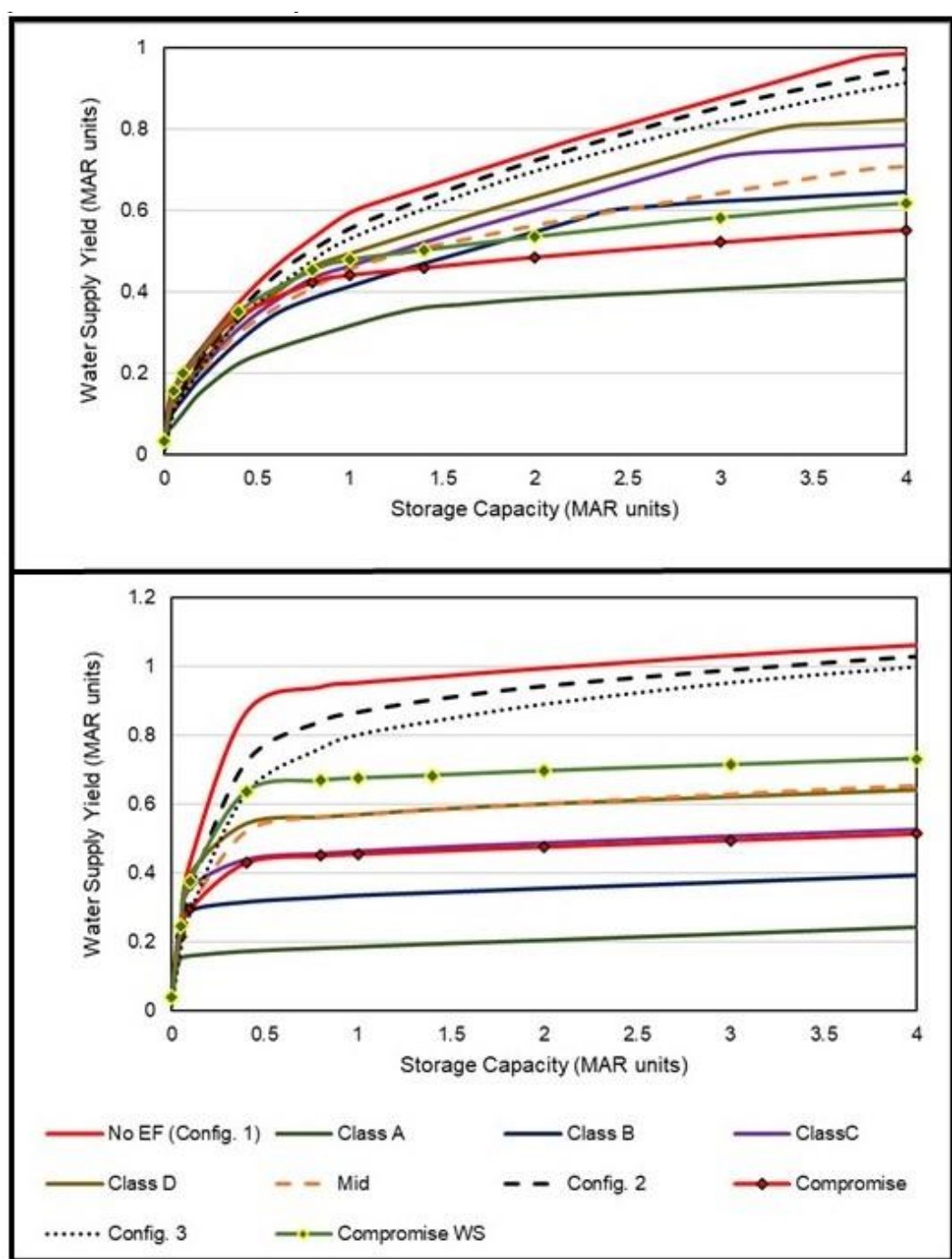

**Figure 7.** Behavior of WS Yield with storage capacity for different Environmental Management Classes, curves of Mid, Compromise, and Compromise WS, and different scenarios of reservoir system configurations for Malwatu Oya (**top**) and Kalu Ganga (**bottom**).

The alignment of the Mid, Compromise, and Compromise WS pathways for the two basins compared to overarching pathways described in Section 3.1 is shown in Figure 7.

In rivers where no specific ecological protection requirement has been identified, an average pathway such as the Mid or Compromise curve (Figure 7) may be adopted. For example, for the Malwatu Oya basin, the Mid and Compromise curves envelop the Environmental Management Class C curve up to a storage capacity of 0.7 MAR units and this envelope moves to the Class B curve beyond a storage capacity of 1.3 MAR units (Figure 7 top). It can be inferred that up to a cumulative storage capacity of approximately 1.0 MAR units, the Class C curve represents a reasonable middle-ground storage devel-

opment pathway unless there is a specific need for the river to be in a more protected state. Figure 7 (top) illustrates that marginal gains in WS Yield from storage capacities above 1.0 MAR units along Mid and Compromise pathways are minimal and that beyond this point accomplishing a satisfactory trade-off between water supply and environmental needs will be challenging. The mean of Mid and Compromise yields at this storage capacity is 45% of the MAR. Both curves level off at an average yield of 65%, corresponding to a storage capacity of 5.0 MAR units. Accordingly, it is possible to define a conservative limit to storage capacity expansion in the Malwatu Oya basin at 1.0 MAR units and 45% of MAR water withdrawals. A maximum withdrawal limit from surface storage may be set at 65% of the MAR, assuming that more stringent levels of ecosystem protection are not necessary.

Similarly, in the case of Kalu Ganga, considering the alignment of Mid and Compromise curves, either the Class C or the Class D curve may be adopted depending on ecosystem protection requirements. A conservative limit to storage capacity expansion in Kalu Ganga may be set at 0.5 MAR units and 50% of MAR water withdrawals and a maximum limit may be set at 60% of MAR, assuming that more stringent levels of ecosystem protection are not necessary.

### 3.3. Optimal Distribution of Storage Capacity and Withdrawal Thresholds in Sub-Basins

Cumulative WS Yield estimates for the entire basin under reservoir system Configurations 2 and 3 (Section 2.5) revealed that the cumulative yield from the entire basin is maximized when the cumulative storage capacity of the basin is distributed in sub-basins according to their MAR ratios (also observed by Pitman in 1995 [53]). However, the maximum WS Yield achieved from Configuration 1 was higher than the maximum aggregated WS Yield obtained from Configurations 2 and 3 for the same cumulative storage capacity (Figure 7). This illustrates that the maximum achievable WS Yield from a lumped reservoir of a certain capacity is higher than the cumulative yield obtained from a network of distributed reservoirs of the same aggregated capacity. The cumulative yield for the entire basin declines with subsequent levels of reservoir distribution but may stabilize at high levels of distribution since homogeneous regions are likely to have similar (normalized) storage versus WS Yield curves. The results imply that more releases are available in downstream parts of the basin as EF when small reservoirs are distributed in the river network, even if no explicit EF releases are made. Distributing storage capacity within sub-basins according to their MAR ratios and limiting abstractions to an appropriate fraction of the local runoff generated within each sub-basin is one strategy to ensure that cumulative water supply yields are maximized, the river basin as a whole complies with a pre-selected storage development pathway, and corresponding EF releases are maintained. Figures 8 and 9 illustrate how such a strategy may be practically formulated in river basin planning.

Figure 8 shows the feasible storage capacity (S) and withdrawals (Y) in each large sub-basin, for the entire Malwatu Oya basin to stay in the Mid scenario if the cumulative storage capacity in the basin were to be increased to 1 MAR unit from its existing capacity of 0.43 MAR units. Figure 9 shows the same distribution for Kalu Ganga if the storage capacity in the basin were increased to 0.5 MAR units. The figures have been produced by distributing the cumulative storage capacity (1.0 MAR units in Malwatu Oya and 0.5 MAR units in Kalu Ganga) in large sub-basins according to their MAR ratios and estimating individual WS Yields corresponding to the storage capacity assigned to each large sub-basin using Storage-WS Yield curves developed under Configuration 2. To estimate the WS-Yield for each large sub-basin under the Mid scenario, individual WS Yields obtained under Configuration 2 were multiplied by the ratio between the ordinates of the Configuration 2 curve and the Mid curve for the entire basin (Figure 7) at 1.0 MAR units in Malwatu Oya and 0.5 MAR units in Kalu Ganga. Similar spatially disaggregated estimates may be made for other cumulative storage capacities and overarching pathways. It is also possible to adopt variable pathways and corresponding storage capacities and withdrawal thresholds for different sub-basins depending on ecological protection requirements.

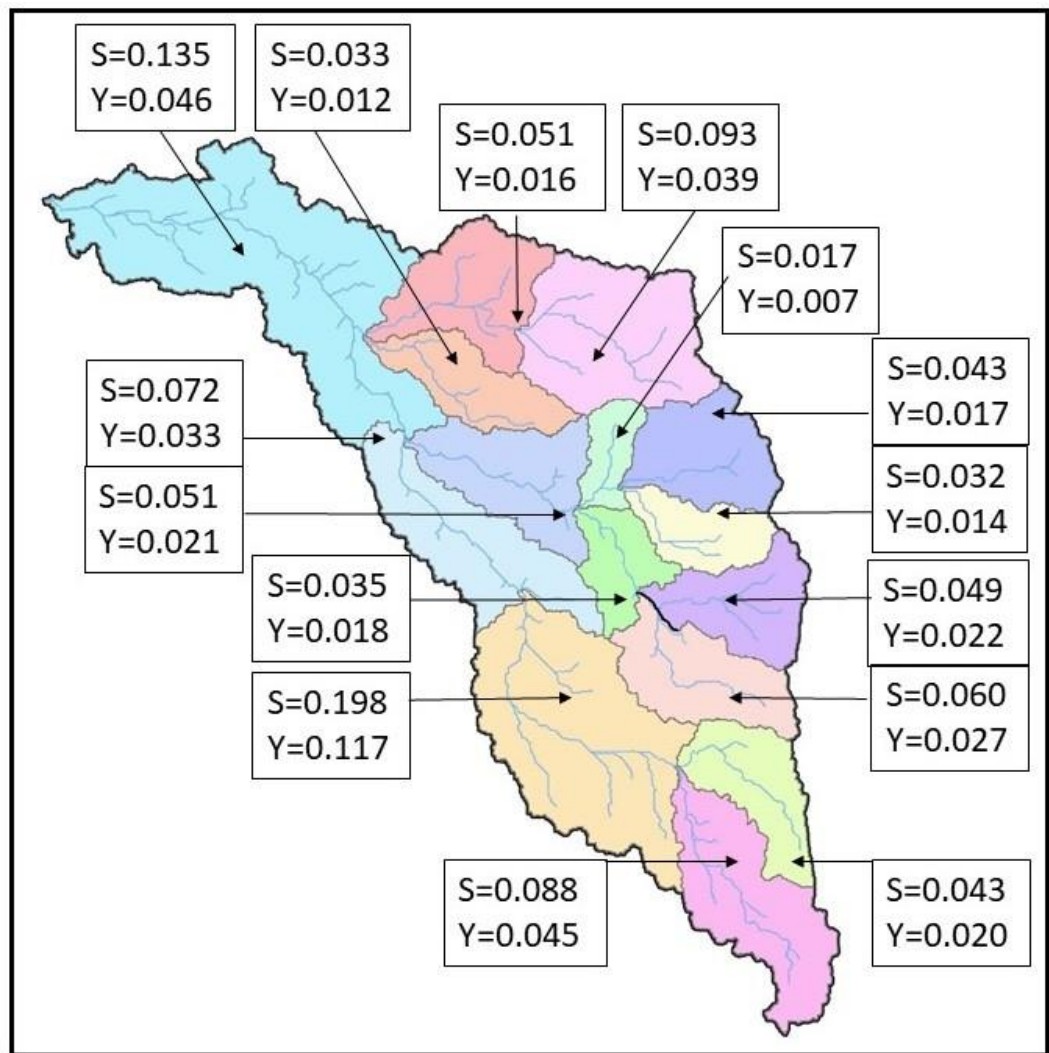

**Figure 8.** Feasible storage capacity (S) and withdrawals (Y) in each large sub-basin of Malwatu Oya for the entire basin to stay on the Mid scenario if the cumulative storage capacity were increased to 1.0 MAR unit. S and Y values given in terms of MAR units of the entire basin.

*3.4. Comparing Results with the Current Situation in the Case Study Basins and Practical Implications for Planning*

The availability of more in-stream flow releases when reservoirs are distributed over the river network than when they are lumped together (Section 3.3) provides a scientific basis for the high level of reservoir distribution seen in the ancient tank irrigation system in the Malwatu Oya basin, although EF is probably not the only reason for this distribution. One aspect not covered in this research is the impact of evaporation on actual WS Yields when storage capacity is distributed over a large number of small reservoirs. Beyond a certain level of disaggregation of reservoir capacity over the river network, losses from evaporation may outweigh the gains in EF. Future research could investigate the relative impact of evaporation versus disaggregation of storage capacity on WS and EF Yields.

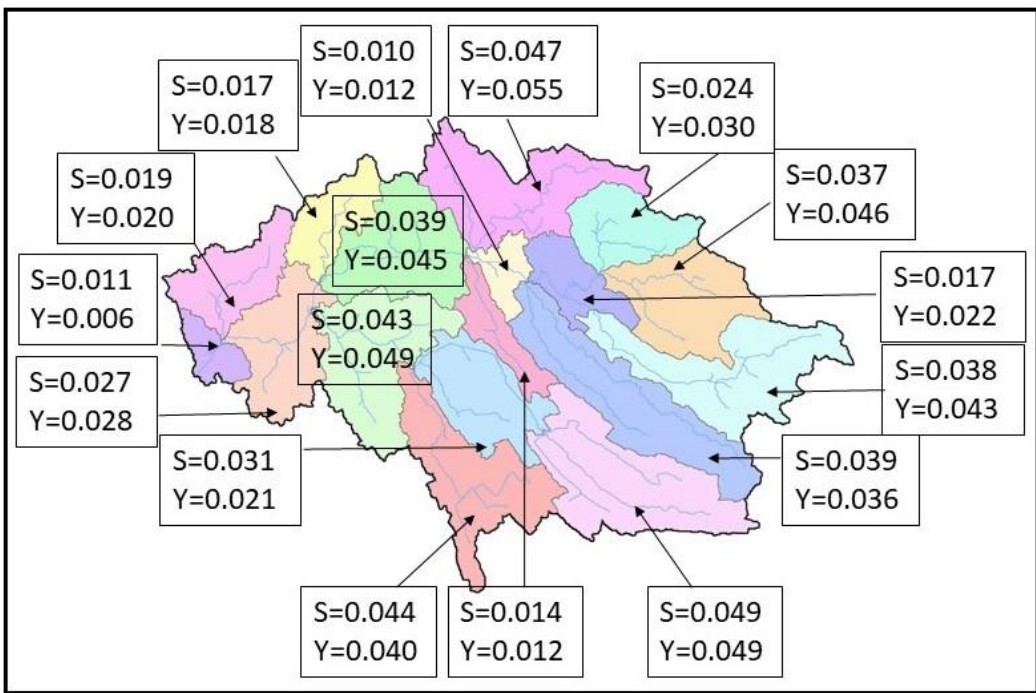

**Figure 9.** Feasible storage capacity (S) and withdrawals (Y) in each large sub-basin of Kalu Ganga for the entire basin to stay on the Mid scenario if the cumulative storage capacity were increased to 0.5 MAR units. S and Y values given in terms of MAR units of the entire basin.

If the existing storage capacities in each minor sub-basin of the Malwatu Oya were lumped together (as in Configuration 3) and were drawing their maximum yields irrespective of EF releases, the cumulative annual 100% temporally reliable WS Yield would be 0.31 of the MAR of Malwatu Oya. Based on a comparison of current basin outflow with estimated natural outflow and actual demands, gross annual withdrawals in the basin were found to be as high as 0.55 MAR units with a time reliability of only 75–80%. The current annual in-stream flow release of 0.45 MAR units implies that the river is in Environmental Management Class B (Table 2). However, a close examination of the actual monthly time series of in-stream flow revealed that although the in-stream flow release is above that for Class B during the wet season, it drops to below that for Class F during the dry season. Hence, the river is in Class F under current practices.

Figure 10 classifies large sub-basins of Malwatu Oya by their current state of exploitation based on a comparison of their actual storage capacity and level of withdrawals with where they should be to place the basin as a whole in the Mid scenario (assuming the Mid scenario is adopted). It illustrates that four of the 15 major sub-basins are over-exploited in terms of storage, while five others are under-exploited. Table 4 compares the current spatial distribution of storage capacity among large sub-basins with the expected distribution to maximize yield gains, i.e., a distribution which follows the ratio among the MAR of large sub-basins. Sub-basins over-/under-exploited by more than 20% are highlighted in Table 4. Positive percentages in the last column imply over-exploitation, while negative percentages imply under-exploitation.

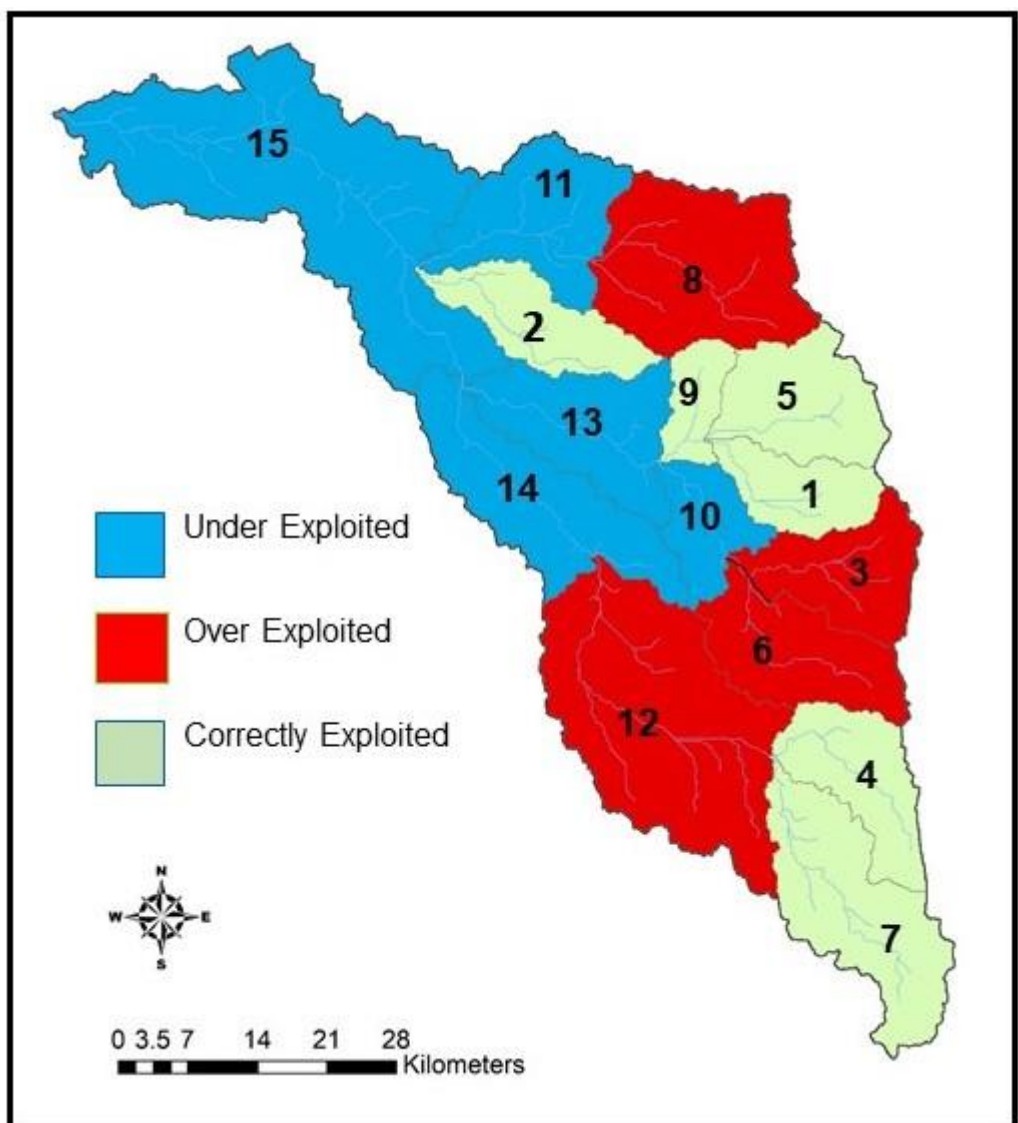

**Figure 10.** Classification of large sub-basins of Malwatu Oya by their current state of exploitation of surface storage. Sub-basin numbers shown in black. Sub-basin details provided in Table 4.

Any new surface water storage development in the Malwatu Oya basin needs to focus on under-exploited areas to maximize yield gains and ensure environmental protection. Since the basin is already saturated with surface storage, the development of sustainable groundwater may be the next best option to release the stress on surface water. In this context, regularization of informal groundwater development currently underway in the basin is a high priority. Any transfers from other basins need to ensure they do not unduly compromise the ecological integrity of the source basins. Since Kalu Ganga is relatively unregulated at present, these results may be useful for development planning.

**Table 4.** Degree of over-/under-exploitation of large sub-basins of Malwatu Oya. Colors blue, red and green correspond to under, over and correct exploitation levels.

| No. | Sub-basin | MAR (mill. m$^3$) | Existing Storage Capacity (mill. m$^3$) | Existing Storage Capacity as % of MAR | MAR as % of the Total Basin MAR | Existing Storage Capacity as % of Total Storage Capacity | Degree of over-/ under-Exploitation |
|---|---|---|---|---|---|---|---|
| 1 | Kadahatu Oya | 26.36 | 10.75 | 40.79 | 3.22 | 3.11 | −3.29 |
| 2 | Narivili Aru | 27.20 | 10.27 | 37.74 | 3.32 | 2.97 | −10.53 |
| 3 | Maha Kanadara Oya | 40.19 | 31.92 | 79.44 | 4.90 | 9.23 | +88.35 |
| 4 | Maminiya Oya | 35.11 | 16.85 | 48.00 | 4.28 | 4.87 | +13.81 |
| 5 | Sangili Kanadara Oya | 35.65 | 17.95 | 50.34 | 4.35 | 5.19 | +19.36 |
| 6 | Upper Kanadara Oya | 49.03 | 35.71 | 72.83 | 5.98 | 10.33 | +72.69 |
| 7 | Upper Malwatu Oya | 71.91 | 35.97 | 50.02 | 8.77 | 10.40 | +18.60 |
| 8 | Boo Oya | 76.35 | 42.83 | 56.09 | 9.31 | 12.39 | +33.00 |
| 9 | Middle Sangili Kanadara Oya | 14.18 | 5.45 | 38.45 | 1.73 | 1.58 | −8.82 |
| 10 | Lower Weli Oya | 28.32 | 2.77 | 9.77 | 3.45 | 0.80 | −76.83 |
| 11 | Kal Aru | 41.67 | 3.28 | 7.86 | 5.08 | 0.95 | −81.35 |
| 12 | Upper Middle Malwatu Oya | 161.94 | 114.24 | 70.54 | 19.75 | 33.04 | +67.26 |
| 13 | Lower Kanadara Oya | 41.96 | 6.65 | 15.84 | 5.12 | 1.92 | −62.43 |
| 14 | Lower Middle Malwatu Oya | 59.37 | 7.26 | 12.23 | 7.24 | 2.10 | −70.99 |
| 15 | Lower Malwatu Oya | 110.56 | 3.86 | 3.49 | 13.49 | 1.12 | −91.72 |
| | Total | 789.70 | 345.75 | 43 | 100 | 100 | |

*3.5. Use of Regionalized Storage-Yield Curves as a Measure of the Surface Storage Potential of River Basins and as a Basin Characteristic*

The shape of the storage-yield curve for a particular river basin depends on the characteristics of the inflow time series, which in turn depend on the climate, geomorphology, and geological features of the basin. For example, the curves for the Malwatu Oya and its sub-basins have a gradual gradient in the initial segment before leveling off, whereas those for Kalu Ganga have a much steeper gradient in the initial segment. Yield gains from storage are reached much quicker in the latter than in the former. The maximum potential yield (No EF) curve for Malwatu Oya levels off at a storage capacity of 3.8 MAR units (3001 million m$^3$), whereas in Kalu Ganga this point is reached much earlier at a capacity of 0.4 MAR units (3381 million m$^3$) (Figure 7). The corresponding yields are 772 and 7312 million m$^3$, respectively. Although the storage capacities at which the maximum yields are reached are of the same order of magnitude, the corresponding yields differ by an order of magnitude 10.

A factor analysis by principal components of the set of curves for the 148 minor sub-basins of the Malwatu Oya revealed that 88% of the variance in the shape of the curves (especially for storage capacities above 0.1 MAR units) may be explained by one common factor, which is highly positively correlated with the mean and standard deviation, and negatively correlated with the skewness of annual precipitation of individual sub-basins. Therefore, 88% of the variance in runoff generated by the minor sub-basins in the Malwatu Oya and their storage-yield curves can be explained by their annual precipitation statistics. Considering the variation of factor values across the basin, Malwatu Oya may be represented by six regional storage-yield curves (Figure 11; Table 5).

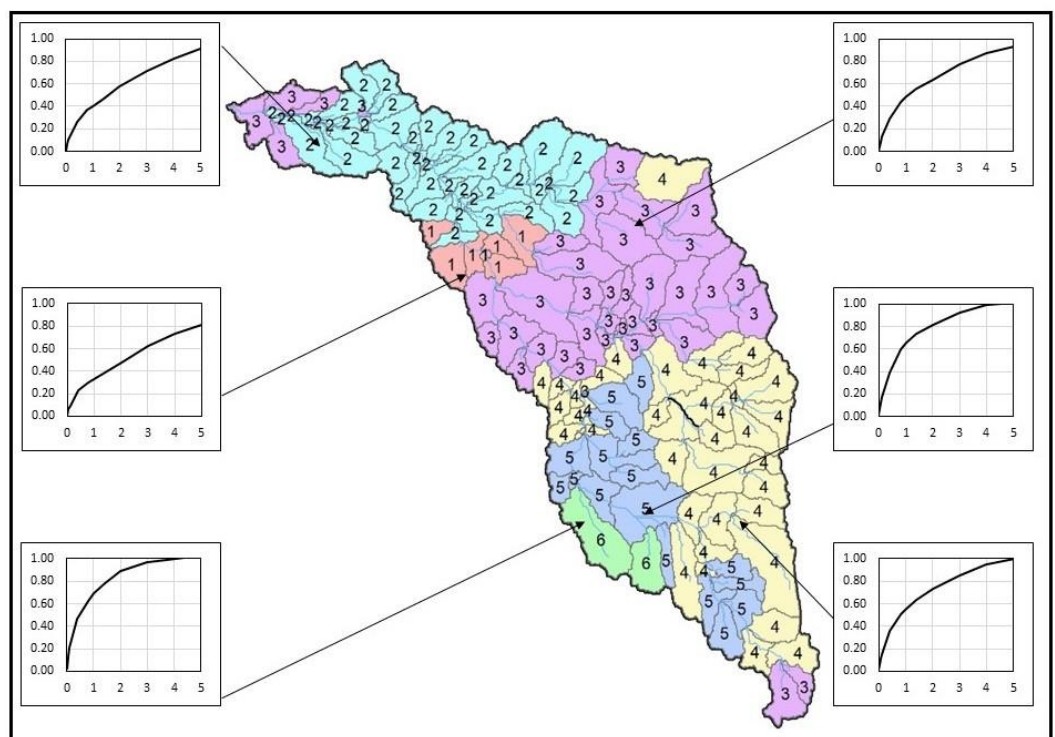

**Figure 11.** Regional Storage-Yield Curves of Malwatu Oya with minor sub-basins having similar curves indicated by color and number. Graphs show storage capacity on the horizontal axis; WS Yield on the vertical axis in MAR units.

**Table 5.** Regional storage-yield curves represented as flow tables for Malwatu Oya and Kalu Ganga basins (Figure 11, Figure 12).

| Storage Capacity (MAR units) | WS Yield (MAR units) | | | | | | | | | | | | | | |
| | Malwatu Oya | | | | | | Kalu Ganga | | | | | | | | |
| | Region Number | | | | | | | | | | | | | | |
| | 1 | 2 | 3 | 4 | 5 | 6 | 1 | 2 | 3 | 4 | 5 | 6 | 7 | 8 | 9 |
| 0.00 | 0.0 | 0.00 | 0.01 | 0.01 | 0.01 | 0.02 | 0.00 | 0.01 | 0.00 | 0.00 | 0.00 | 0.01 | 0.02 | 0.03 | 0.03 |
| 0.05 | 0.07 | 0.07 | 0.09 | 0.11 | 0.12 | 0.13 | 0.06 | 0.16 | 0.08 | 0.16 | 0.17 | 0.19 | 0.22 | 0.23 | 0.23 |
| 0.1 | 0.10 | 0.11 | 0.13 | 0.16 | 0.17 | 0.21 | 0.10 | 0.23 | 0.15 | 0.28 | 0.29 | 0.31 | 0.33 | 0.37 | 0.40 |
| 0.4 | 0.23 | 0.26 | 0.30 | 0.36 | 0.39 | 0.46 | 0.22 | 0.37 | 0.42 | 0.67 | 0.63 | 0.66 | 0.74 | 0.76 | 0.84 |
| 0.8 | 0.30 | 0.37 | 0.44 | 0.51 | 0.60 | 0.63 | 0.30 | 0.48 | 0.69 | 0.85 | 0.78 | 0.75 | 0.81 | 0.88 | 0.94 |
| 1.0 | 0.33 | 0.40 | 0.49 | 0.56 | 0.66 | 0.69 | 0.33 | 0.54 | 0.80 | 0.92 | 0.83 | 0.78 | 0.85 | 0.90 | 0.95 |
| 1.4 | 0.39 | 0.47 | 0.55 | 0.63 | 0.73 | 0.78 | 0.41 | 0.65 | 0.84 | 0.96 | 0.89 | 0.81 | 0.86 | 0.93 | 0.97 |
| 2 | 0.48 | 0.58 | 0.64 | 0.73 | 0.82 | 0.89 | 0.52 | 0.82 | 0.89 | 0.99 | 0.97 | 0.87 | 0.88 | 0.96 | 0.99 |
| 3 | 0.62 | 0.71 | 0.78 | 0.85 | 0.92 | 0.97 | 0.67 | 0.95 | 0.95 | 1.02 | 1.00 | 0.94 | 0.91 | 1.00 | 1.01 |
| 4 | 0.73 | 0.83 | 0.87 | 0.95 | 0.99 | 1.00 | 0.76 | 1.00 | 1.00 | 1.05 | 1.03 | 1.00 | 0.94 | 1.03 | 1.04 |
| 5 | 0.81 | 0.91 | 0.93 | 1.00 | 1.01 | 1.03 | 0.83 | 1.04 | 1.03 | 1.07 | 1.05 | 1.04 | 0.97 | 1.06 | 1.07 |

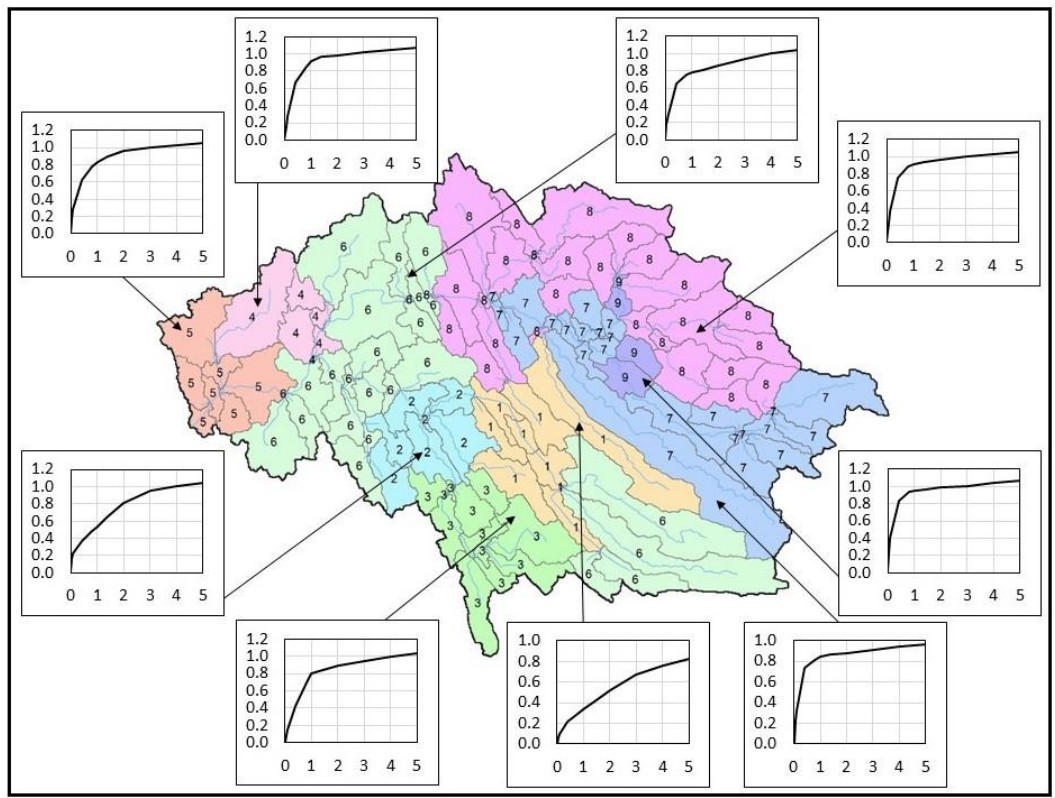

**Figure 12.** Regional Storage-Yield Curves of Kalu Ganga with minor sub-basins having similar curves indicated by color and number. Graphs show storage capacity on the horizontal axis; WS Yield on the vertical axis in MAR units.

In the Kalu Ganga, 92% of the variance in the curves could be explained by two common factors, with the first factor accounting for 77%. The first factor explains the variance in the flatter part of the curves beyond 0.4 MAR units, and the second explains the variance in the initial gradient up to 0.4 MAR units. Variance in the flatter part of the curves beyond 0.4 MAR units (explained by factor 1) is negatively correlated with the standard deviation, and positively correlated with the skewness of annual precipitation. It was also found that variance in the initial gradient of the curves up to 0.4 MAR units (explained by factor 2) is positively correlated with the mean annual precipitation. Therefore, 92% of the variance in runoff generated by individual minor sub-basins in the Kalu Ganga and their storage-yield curves can be explained by their annual precipitation statistics. Considering the variation of the common factors across the basin, Kalu Ganga is represented by nine regional storage-yield curves (Figure 12, Table 5).

The differences in storage-yield curves among basins and sub-basins within each basin can be explained by differences in the statistics of annual precipitation. However, since simulated runoff was used, the characteristics of the storage-yield curves depend on the climate and catchment characteristics input during modeling. Hence, the actual extent to which storage yield curves are influenced by characteristics of annual precipitation versus other catchment characteristics needs to be verified at places where substantial measured flow data are available.

According to the generalized global storage–yield–reliability relationship developed by Kuria and Vogel [54,55], standardized yields (yields as a fraction of the MAR) in the range 0.3–0.8 from a given reservoir capacity can be expressed by a combination of the statistics of the inflow time series (a combination of mean, standard deviation, and skewness coefficient of annual inflows; Equation (9)). The possibility of replacing the statistics of annual inflows with the statistics of annual precipitation needs to be further explored and would be beneficial for storage planning in regions where measured and modeled runoff data are not readily available. Emerging methods for generating synthetic flow records

such as continuous simulation modeling (e.g., [56,57]) may also prove useful to establish storage-yield curves in ungauged basins.

$$Y = 0.651 S^{0.203} Z_r^{-0.306} \mu^{1.135} \sigma^{-0.342} \gamma^{0.017} \tag{9}$$

where $Y$ = yield; $S$ = storage capacity in million cubic metres; $Z_r$ = standard normal variate with r equal to reliability; $\mu, \sigma, \gamma$ = the mean, standard deviation, and skewness coefficient of annual inflows.

## 4. Conclusions

The results of this research illustrate that Storage-WS yield curves for entire river basins under different EF release scenarios are a useful approach for designing overarching sustainable storage development pathways and acceptable limits to surface storage development in river basins. Storage-WS Yield curves and the indices of WS Sustainability and EF Sustainability may be conjunctively used to identify the most appropriate storage development pathway and acceptable storage limits for a given river basin so that aquatic ecosystem health and related services are not unduly compromised as the basin's cumulative storage capacity increases. The research demonstrates that the optimal arrangement of storage capacity is to have the cumulative capacity distributed among sub-basins according to the MAR ratio among them, and that withdrawals from each sub-basin may be appropriately limited to maintain the entire basin on the identified pathway.

While this approach may help plan long-term storage development in relatively underdeveloped river basins, it may also be used to compare the current status of developed basins against what is acceptable, and on that basis to propose remedies to any deviations. Alternative approaches for establishing storage-yield curves including the use of synthetic flow records may prove a fruitful future direction in research to implement this paper's recommendations in regions with limited river flow data.

The research fills an important knowledge gap on the questions of acceptable storage limits and optimal reservoir arrangements for river basins. This generic approach can be adapted to individual river basins by: (1) using reliabilities of supply less than 100%; (2) using alternative EF estimation methods; (3) formulating an alternative EF Sustainability index, measuring the extent to which instream flow releases differ from natural flows, instead of its current formulation; (4) considering evaporation and percolation losses and inactive storage zones for reservoirs; (5) using daily flow data instead of monthly flow data especially in the case of smaller capacity reservoirs; (6) including indicators on economic sustainability to compare different reservoir system configurations; (7) expanding the approach to cover other water use sectors including non-consumptive ones and, (8) considering future climate and socioeconomic scenarios.

**Author Contributions:** Conceptualization, N.E., V.S. and L.U.; methodology, N.E.; formal analysis, N.E.; investigation, N.E.; writing—original draft preparation, N.E.; writing—review & editing, V.S. and L.U.; supervision, V.S. and L.U. All authors have approved the content of the submitted manuscript.

**Funding:** This research was supported by the International Water Management Institute as part of the CGIAR Research Programs on Climate Change Agriculture and Food Security and Water Land and Ecosystems.

**Institutional Review Board Statement:** Not applicable.

**Informed Consent Statement:** Not applicable.

**Data Availability Statement:** Simulated runoff data used in the analysis may be made available on request from the authors.

**Acknowledgments:** The authors are grateful to Matthew McCartney, Research Group Leader, Sustainable Water Infrastructure and Ecosystems (IWMI) for his constructive feedback on the manuscript and would also like to thank Lal Mutuwatte (IWMI) and Madusanka Thilakarathne (Consultant, IWMI) for their help in developing the SWAT models.

**Conflicts of Interest:** The authors declare no conflict of interest.

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
