# Peer review of "Sustainable Surface Water Storage Development Pathways and Acceptable Limits for River Basins"

_water, doi:10.3390/w13050645_

Round 1

Reviewer 1 Report

Main text should be shorter, reader will be tired after read whole of this paper. It is a bit long.

Abstract needs some numerical result.

Introduction must be improved.

Use the better form for Figure 4. Please draw it more professionally.

Discussion section is missed! Add it. And also, add some justification in discussion part.

Reviewer 2 Report

A BRIEF SUMMARY

The paper titled ““Sustainable Surface Water Storage Development Pathways and Acceptable Limits for River Basins” presents an interesting topic for readers of this Journal. In the last decade, this issue represents a line of research very investigated. In general the paper provide an original contribution, therefore I invite the authors to go ahead.

Hovewer, some minor questions remain after reading the paper. Below is the list of some questions that need to be addressed.

  • Why the references is so few? Probably a more accurate references research could help to add value for this topic. I strongly suggest that the authors try to add some more references especially in the "part 1 (introduction)" of the paper.
  • In conclusion, the authors as a future improvement of this work can add the use of “continuous hydrological model”. I have indicated suggestions, but more could be added to make the foundation for the arguments stronger.

  • Rowe, T.J., Smithers, J.C., 2018. Continuous simulation modelling for design flood estimation - a South African perspective and recommendations. Water SA 44 (4), 691–705.
  • Winter, B., Schneeberger, K., Dung, N.V., Huttenlau, M., Achleitner, S., St¨otter, J., Merz, B., Vorogushyn, S., 2019. A continuous modelling approach for design flood estimation on sub-daily time scale. Sci. J. 64 (5), 539–554.
  • Grimaldi, S., Nardi, F., Piscopia, R., Petroselli, A., Apollonio, C.. Continuous hydrologic modelling for design simulation in small and ungauged basins: A step forward and some tests for its practical use. Journal of Hydrology, 2020, 125664, ISSN 0022-1694. doi: 10.1016/j.jhydrol.2020.125664

SPECIFIC COMMENTS:

Figure 4 and Figure 5: I suggest replacing these figures after a quality improvement.

Reviewer 3 Report

I have read with interest your paper on Sustainable Surface Water Storage Development Pathways and Acceptable Limits for River Basins. I have found it well written, clear, and well substantiated. I also enjoyed the fact it is well placed within the relevant literature, as shown in the introduction. 

My suggestions for improvement, however, are the following:

  • Figure 1 is not fully within the paper, so please pay attention to its formatting
  • The second paragraph of the introduction on the issue of dams is very interesting, and it would benefit from engaging with and inclusion of, the following literature: Poff, N. L., et al. (2017). Can dams be designed for sustainability?. Science358(6368), 1252-1253; Beck, M. W., et al. (2012). Environmental and livelihood impacts of dams: common lessons across development gradients that challenge sustainability. International journal of river basin management10(1), 73-92; Conker, A., et al. (2020). Small is beautiful but not trendy: Understanding the allure of big hydraulic works in the Euphrates-Tigris and Nile waterscapes. Mediterranean Politics, 1-24; Conker, A., et al. (2019). Hydraulic mission at home, hydraulic mission abroad? Examining Turkey’s regional ‘pax-aquarum’and its limits. Sustainability11(1), 228; Menga, F. (2015). Building a nation through a dam: the case of Rogun in Tajikistan. Nationalities Papers43(3), 479-494.; Conker, A., et al. (2020). Hydropolitics and issue-linkage along the Orontes River Basin: an analysis of the Lebanon–Syria and Syria–Turkey hydropolitical relations. International Environmental Agreements: Politics, Law and Economics20(1), 103-121.
  • Better explain how the scenarios have been created and generated;
  • Formatting on page 11 of the formulas table
  • All figures seem to be quite cut off on the right side
  • Discussion should engage more with the guiding research question, the gap in the literature, and with the frames used.

I hope these suggestions are useful in improving this paper.

Round 2

Reviewer 1 Report

Literature review requires more newly published papers, then improve literature review by new publications.

Reviewer 2 Report

The paper has been improved following reviewer comments. Keep attention, you have to update the reference paragraph with the new citations.

In my opinion, it is ready for publication. Congratulations.

Reviewer 3 Report

Thanks for revising this paper. However, I am not satisfied yet with how the Introduction has been reworked. In fact, I can see that only the references of a certain strain of literature (that I have also mentioned) have been incorporated, but not those by Turkish and Italian colleagues, which are very well known in the British academia. I would therefore suggest, if possible, for the authors to familiarise with the references mentioned earlier of Prof. Filippo Menga, Prof. Ahmet Conker, et al. so that the paper can provide a better engagement with the relevant literature. 

Concerning being concise and word limit, these edits would not need more than 300 words, and Water - MDPI does not have a word limit, so the journal prefers a well done and well situated paper rather than a brief and weak one. 

I hope however, my comments are useful to better situate the paper. I hope my comments are precise enough in order to allow you a fast and not time consuming revision. 

Round 3

Reviewer 1 Report

This manuscript is acceptable.

Reviewer 3 Report

This version looks much improved and more inclusive and comprehensive, well done. 
the reference list should not be in alphabetical other, but I’m sure MDPI will support you with further guidelines during production.